# Pilot RNAi Screen in *Drosophila* Neural Stem Cell Lineages to Identify Novel Tumor Suppressor Genes Involved in Asymmetric Cell Division

**DOI:** 10.3390/ijms222111332

**Published:** 2021-10-20

**Authors:** Sandra Manzanero-Ortiz, Ana de Torres-Jurado, Rubí Hernández-Rojas, Ana Carmena

**Affiliations:** Developmental Neurobiology Department, Instituto de Neurociencias, Consejo Superior de Investigaciones Científicas/Universidad Miguel Hernández, 03550 Sant Joan d’Alacant, Alicante, Spain; smanzanero@umh.es (S.M.-O.); ana.torrese@umh.es (A.d.T.-J.); rubi.hernandezr@umh.es (R.H.-R.)

**Keywords:** asymmetric cell division, tumorigenesis, neural stem cell, *Ras^V^*^12^ *scribble*, *RNAi* screen, tumor suppressor genes, *Drosophila*

## Abstract

A connection between compromised asymmetric cell division (ACD) and tumorigenesis was proven some years ago using *Drosophila* larval brain neural stem cells, called neuroblasts (NBs), as a model system. Since then, we have learned that compromised ACD does not always promote tumorigenesis, as ACD is an extremely well-regulated process in which redundancy substantially overcomes potential ACD failures. Considering this, we have performed a pilot RNAi screen in *Drosophila* larval brain NB lineages using *Ras^V^*^12^ *scribble (scrib)* mutant clones as a sensitized genetic background, in which ACD is affected but does not cause tumoral growth. First, as a proof of concept, we have tested known ACD regulators in this sensitized background, such as *lethal (2) giant larvae* and *warts.* Although the downregulation of these ACD modulators in NB clones does not induce tumorigenesis, their downregulation along with *Ras^V^*^12^ *scrib* does cause tumor-like overgrowth. Based on these results, we have randomly screened 79 RNAi lines detecting 15 potential novel ACD regulators/tumor suppressor genes. We conclude that *Ras^V^*^12^ *scrib* is a good sensitized genetic background in which to identify tumor suppressor genes involved in NB ACD, whose function could otherwise be masked by the high redundancy of the ACD process.

## 1. Introduction

Asymmetric cell division (ACD) is an effective and conserved strategy to generate cell diversity, an issue especially relevant during the development of the nervous system [1,2,3,4]. The neural stem cells of the *Drosophila* central nervous system, called neuroblasts (NBs), divide asymmetrically and have been used as a paradigm for analyzing this process for a long time [5,6,7]. In ACD, one daughter cell keeps on proliferating while its sibling is committed to initiating a process of differentiation. NB asymmetric division renders another self-renewal NB and a daughter cell called a ganglion mother cell (GMC), which will divide once more asymmetrically to give rise to two neurons or glial cells. A group of proteins called cell-fate determinants, such as the cytoplasmic protein Numb, the transcription factor Prospero (Pros)/Prox 1 in vertebrates, and the translational regulator brain tumor (Brat)/TRIM3, accumulate asymmetrically at the basal pole of the NB; then, when the NB divides only the most basal cell, the GMC receives those determinants, which inhibit proliferation and induce differentiation in this cell [8,9,10,11,12,13,14,15,16]. The asymmetric localization of cell-fate determinants, as well as the correct orientation of the mitotic spindle along the apical–basal axis of cell polarity, is, in turn, tightly regulated by multiple proteins, sometimes acting redundantly, to finally ensure the correct asymmetry of the division. For example, a group of proteins enriched at the apical pole of the NB at metaphase (“the apical complex”) that include the plasma-membrane-located GTPase Cdc42, the conserved Par proteins Par6 and Par3 (Bazooka, Baz, in *Drosophila*) and aPKC contribute to excluding the determinants from the apical pole [17,18,19,20,21,22]. These proteins bind the adaptor protein Inscuteable (Insc), which in turn binds Pins/LGN promoting the interaction of Pins/LGN with the Gαi subunit anchored to the membrane [23,24,25,26,27,28,29,30]. Then, the actin-binding protein Canoe (Cno)/Afadin displaces Insc to bind Pins/LGN, fostering the recruitment of Dlg1-Khc73 and the microtubule binding protein Mushroom body defect Mud/NuMA to Pins/LGN and, consequently, the orchestration of the mitotic spindle orientation along the apical–basal axis [31,32,33,34,35,36].

Intriguingly, a connection between failures in the process of ACD and tumorigenesis was demonstrated about 15 years ago using as a model system the NBs of the *Drosophila* larval brain [37]. In this work, pieces of GFP-labeled mutant brains for genes that regulate ACD were able to induce tumor-like overgrowth after being transplanted into the abdomen of wild-type (wt) adult fly hosts [37]. Remarkably, *Drosophila* genes originally identified as tumor suppressor genes, such as *discs large1* (*dlg1*)/*DLG1*, *lethal (2) giant larvae* (*l(2)gl*)/*LLGL1*, and *brain tumor* (*brat*)/*TRIM3* were shown a posteriori to be key regulators of ACD [38,39,40], further supporting the link between failures in ACD and tumorigenesis. Nevertheless, not all ACD regulators lead to tumor-like overgrowth when they are compromised [41]. In a recent work, we observed that NB mutant clones in the larval brain for the ACD regulators Cno/Afadin, Scribble (Scrib), L(2)gl/Llgl1 or Dlg1 do not cause tumor-like overgrowth, although all mutant clones show ectopic NBs [42]. In this study, we used the type II NB lineages (NBII) as a model system, in which the NB divides to give rise to another NB and, instead of a GMC, a progenitor cell called an intermediate neural progenitor (INP) that undertakes an additional round of division to generate another INP and a GMC (Figure 1a) [10,43,44]. Thus, given this extra phase of proliferation, these NBII lineages are more prone to induce tumor-like overgrowth when the process of ACD fails. Given that ACD is a highly redundant process, we reasoned that it would be necessary to downregulate more than one ACD regulator to observe more drastic effects. In fact, we showed that *cno scrib* double-mutant NB clones do display tumor-like overgrowth. Intriguingly, this phenotype is the consequence of losing two ACD regulators, but also of Ras upregulation after evading Cno-mediated repression [43]. In fact, the downregulation of Ras in *cno scrib* NBII clones is enough to suppress the tumor-like overgrowth observed in the double-mutant clones [42]. In addition, overexpressing an activated form of Ras (Ras^V12^) in *scrib* NBII clones is not sufficient to induce those tumoral overgrowths observed in *cno scrib* double mutant clones, even though Ras^V12^ is able to rescue the JNK-mediated apoptosis induced in *scrib* NB mutant clones [42]. With all these results, we hypothesized that *Ras^V^*^12^
*scrib* mutant clones could be an excellent sensitized genetic background in which to screen for novel tumor suppressor genes and potential ACD regulators. Here, we show results that validate that hypothesis and a pilot screen to determine the suitability and the efficiency of the process in this search.

## 2. Results and Discussion

### 2.1. Ras^V12^ scrib NBII Mutant Clones Do Not Show Tumor-like Overgrowth

In *Drosophila* epithelial tissues, oncogenic Ras (Ras^V12^) induces neoplastic overgrowth in combination with cell polarity genes, including *scrib* [45,46]. However, our previous results showed that *Ras^V^*^12^
*scrib* NBII clones survive and show ectopic NBs, but they do not display massive overgrowth [42]. Thus, we reasoned that we could use this sensitized genetic mutant background to screen for novel tumor suppressor genes required in ACD. With this aim, we wanted first to analyze the *Ras^V^*^12^
*scrib* double-mutant phenotype in detail. Following our previous work, we focused this analysis on NBII lineages (Figure 1a). In these NBII clones, the transcription factor Deadpan (Dpn) labels all progenitor cells (the stem-like NB and the mature INPs), whereas the transcription factor Asense (Ase) is only expressed in the INPs (Figure 1a). We observed that *scrib* null mutant clones appeared with low frequency (in 5 brains out of 19) and were of small size compared with control clones (Figure 1b). However, *Ras^V^*^12^
*scrib* NBII clones were detected at the same frequency (in 17 brains out of 30) as control clones and their size was variable. Most of the *Ras^V^*^12^
*scrib* NBII clones were smaller than control clones, with some ectopic NBs and appearing frequently in groups; some were similar to control clones and few of them were composed mainly of NBs, but none of them show tumor-like overgrowth (Figure 1c).

### 2.2. Downregulation of Known ACD Regulators in Ras^V12^ scrib NBII Clones Induces Tumor-Like Overgrowth

Based on the *Ras^V^*^12^
*scrib* NBII mutant clone phenotype, we inferred that the tumor-like overgrowth observed in *cno scrib* null mutant NBII clones [42] was induced not just by the upregulation of Ras, caused by the absence of its inhibitor Cno, but also by the simultaneous loss of two ACD regulators, Cno and Scrib. This would imply that we could search for novel ACD regulators, whose loss along with the *Ras^V^*^12^
*scrib* condition could induce tumor-like overgrowth. To prove this hypothesis, we first performed a qualitative inquiry approach testing known ACD regulators. We started looking at *dlg1* and *l(2)gl*, as we had observed that the downregulation of each of them in NBII clones does not cause tumoral growth [42]. Intriguingly, we observed some brains with *dlg1^RNAi^; Ras^V^*^12^
*scrib* NBII clones bigger than *Ras^V^*^12^
*scrib* clones and filled mainly by NBs (Dpn^+^ Ase^−^) (Figure 2a), a phenotype that also appeared and was much more exacerbated in brains with *l(2)gl^RNAi^; Ras^V^*^12^
*scrib* NBII clones, which expanded in some cases throughout the brain hemisphere (Figure 2b). Previously, we described a novel function of Warts (Wts), a core component of the Hippo tumor suppressor signaling pathway, in ACD, phosphorylating and stabilizing Cno/Afadin at the apical pole of mitotic NBs. However, as in the case of *l(2)gl* and *dlg1*, *wts^x^*^1^ NBII single-mutant clones do not show tumor-like overgrowth [32]. Hence, we looked at the effect of downregulating *wts* along with *Ras^V^*^12^
*scrib* observing big *wts^RNAi^; Ras^V^*^12^
*scrib* NBII clones showing tumor-like overgrowth. (Figure 2c). In conclusion, the above results strongly supported the reasoning of our hypothesis to find novel ACD regulators/tumor suppressor genes and, based on that, we decided to design and carry out a pilot screen to further prove it.

### 2.3. Screen Outline and Controls

A total of 79 second chromosome *UAS-RNAi* lines from Vienna *Drosophila* Resource Center (VDRC) GD or KK collections were randomly screened. Those *UAS-RNAi* lines were combined with *UAS-Ras^V^*^12^
*FRT82B scrib (Ras^V^*^12^
*scrib)* on the third chromosome to perform MARCM clones [44] and to search under the fluorescence microscope for NBII clones with tumor-like overgrowth (TLO from hereon) (Figure 3). To facilitate the analysis and identification of potential “positive” *UAS-RNAi* lines among the screened lines, different controls were first run. For example, to clearly identify larval brains with GFP clones, instead of any leaky GFP expression, we carried out a “background” control, in which recombination of the Gal4 repressor Gal80 is not taking place; thus, Gal4 cannot drive the expression of *UAS-CD8::GFP* and any GFP detected would correspond to leaky GFP or autofluorescence (Figure 4a and Table 1). In addition, a negative control consisting of *Ras^V^*^12^
*scrib* mutant clones, without any *UAS-RNAi* line on the second chromosome, was also taken into account. A total of 35 larvae with *Ras^V^*^12^
*scrib* clones were analyzed to define the biggest *Ras^V^*^12^
*scrib* clones we were able to detect (Figure 4b and Table 1). Thus, any experimental line showing mutant clones similar to those would be considered negative, whereas those mutant clones clearly above that size would be classified as lines with TLO and potential “positive” lines. Finally, as positive controls, we included the *UAS-RNAi* lines of *l(2)gl*, *dlg1*, and *wts*, which were analyzed following the scheme of the screening (Figure 3 and Table 1). We could unambiguously detect a significant percentage of *l(2)gl^RNAi^*; *Ras^V^*^12^
*scrib* and *wts^RNAi^; Ras^V^*^12^
*scrib* larvae showing brains with TLO (Figure 4c and Table 1). However, under the conditions of the screen, we were not able to detect clear cases of TLO in *dlg1^RNAi^*; *Ras^V^*^12^
*scrib* larval brains (Figure 4c and Table 1). We already noticed in the “proof of concept” experiment, the staining with Dpn/Ase (see above), that the expressivity and penetrance of the *dlg1^RNAi^*; *Ras^V^*^12^
*scrib* phenotype was lower than in *l(2)gl^RNAi^*; *Ras^V^*^12^
*scrib* or than in *wts^RNAi^; Ras^V^*^12^
*scrib* mutant combinations. In addition, under the screen conditions, Dpn/Ase markers, which helped to identify tumoral masses in the brain filled with NBs, stem-like cells, are not present. The fact that we were not able to detect clear cases of TLO in *dlg1^RNAi^*; *Ras^V^*^12^
*scrib* larval brains indicated that we were probably going to miss some potential candidates (ACD regulators) that behave similarly to *dlg1*. Nevertheless, the evident cases of TLO found in the other positive controls, *l(2)gl* and *wts*, ensured the identification of those potential ACD regulators that display such strong interactions with *Ras^V^*^12^
*scrib* as *l(2)gl* and *wts* do.

### 2.4. Positive UAS-RNAi Lines

Once we established all the controls, we started to randomly screen the “experimental” *UAS-RNAi* lines. Seventy-nine *UAS-RNAi* lines on the second chromosome were analyzed in combination with *Ras^V^*^12^
*scrib*. At least 12 larvae with clones from each line were observed under the microscope. We decided that those lines in which TLO clones were not detected after analyzing 12 larvae would be directly classified as “negative”. In addition, we considered that at least 2 larvae with evident cases of TLO clones should be detected to establish the line as a “positive”. Thus, those lines in which only 1 TLO was observed after analyzing 12 larvae were further screened (until a maximum of 30 larvae) looking for at least another case of clear TLO to confirm the line as positive (Table 1). After finishing the screen, we had identified 15 potential positive lines (Figure 5 and Table 1).

Intriguingly, among those potential positive lines, we detected known ACD regulators, such as line 9, RNAi corresponding to the gene *14-3-3-ζ*, which encodes a protein that participates in the proper orientation of the mitotic spindle in dividing NBs [47]. Another positive line, line 65, was identified as an *enhancer of yellow 3*, *e(y)3*, which encodes a nuclear protein that physically and functionally interacts with both the transcription initiation factor TFIID and the SWI/SNF chromatin remodeling complex [48,49]. This complex is key to preventing tumorigenesis within *Drosophila* larval brain neural lineages by avoiding the de-differentiation of intermediate neural progenitors to an NB, stem-like cell fate [50]. Hence, the identification of these lines supports the suitability of the screen to identify novel ACD regulators.

### 2.5. Analysis of the UAS-RNAi Line 68

To further validate the screen, we decided to select the line that showed the highest percent of TLO cases without showing any off-target effects, the line 68, to perform additional analyses. This line was identified as *Actin-related protein 8 (Arp8)*, which encodes a proposed core component of the chromatin remodeling INO80 complex (Flybase). First, we determined the size of the selected *UAS-RNAi* line single-mutant clone; this was to discard the possibility that the TLO phenotype observed in the *UAS-RNAi; Ras^V^*^12^
*scrib* combination was just due to the downregulation of the gene associated with the line (that, otherwise, would also be interesting). The downregulation of the gene associated with that line in NBII lineages did not show TLO by itself in any of the larvae examined (*n* = 15; Figure 6a). Then, we analyzed the phenotype of the selected *UAS-RNAi line* in NBII clones, looking for defects in the ACD process. Specifically, we searched for potential failures in the localization of two ACD regulators, the apical protein aPKC and the cell fate determinant Numb, in dividing progenitors within the clone. Although no significant defects in the localization of Numb were observed, we detected significant failures in the localization of the apical protein aPKC in metaphase progenitors (Figure 6b). Thus, although it will be required to perform further and detailed analyses in the future, these results already suggest that Arp8 somehow contributes to the regulation of ACD, and that other “positive lines” might also represent known or novel ACD modulators.

## 3. Conclusions

The pilot screen presented here was performed at a low scale and, therefore, the number of positive lines identified are not yet enough to establish further relationships among them in the context of gene ontology (GO) terms and other similar parameters, an enrichment analysis that could be made in a more robust way on the results of a screen carried out at a higher scale. Nevertheless, this pilot screen strongly supports the hypothesis on which it was based. Likewise, the identification of known ACD regulators, as well as the validation of some of the positive lines, already show that we can isolate novel tumor suppressor genes involved in regulating ACD. Similarly, as we had predicted, we can miss some ACD regulators in this type of screen, as has been the case, for example, of the apical protein Par-3/Bazooka, which was found among the “negative” lines. Finally, the high percentage of positive lines identified was unexpected. Hence, additional analyses will be carried out in all those lines; this will further validate and confirm the capability of this screen to uncover novel regulators and mechanisms involved in ACD modulation.

## 4. Materials and Methods

### 4.1. Drosophila Strains and Genetics

The fly stocks used were from the Bloomington *Drosophila* Stock Center (BDSC) and the Vienna *Drosophila* Resource Center (VDRC), unless otherwise stated: *hs-FLP* (X chromosome); *UAS-Ras^V^*^12^
*FRT82B; UAS-Ras^V^*^12^
*FRT82B scrib*^2^*; FRT82B scrib*^2^ (all from H. Richardson); *FRT82B scrib*^1^ (both *scrib*^1^ and *scrib*^2^ are null alleles [45,51,52] *FRT82B; Dll-Gal4 UAS-CD8::GFP*; *FRT82B tub-Gal80*; *UAS-CD8::GFP*; *wor-Gal4 ase-Gal80* [53]; *UAS-l(2)gl^RNAi^* (VDRC: 109604); *UAS-dlg1^RNAi^* (VDRC: 41134); *UAS-wts^RNAi^* (VDRC: 106174); *40D-UAS* (control for KK library landing site at 40D; VDRC: 60101); all the 79 *UAS-RNAi* lines screened were lines on the second chromosome from the GD or the KK VDRC collection. These lines were randomly selected from a big UAS-RNAi collection belonging to M. Domínguez, who kindly let us pick the 79 lines used in this screen. We knew nothing a priori about the identity of the genes; the only requisite we followed was that the lines were on the second chromosome because of the design of the screen (Figure 3).

### 4.2. Histology, Immunofluorescence, and Microscopy

To analyze the *UAS-RNAi* lines of the screen, late L3 larval brains were dissected, mounted without fixation, and analyzed under a Carl Zeiss microscope (Axio Imager.A1), EC Plan-Neofluar 20× objective (Figure 4, Figure 5 and Figure 6a) and an AxioCam Hrc Carl Zeiss camera. Images were assembled using Adobe Photoshop CS6.

To perform the immunofluorescence, L3 larval brains were dissected in PBS and fixed with 4% PFA in PBT (PBS and Triton X-100 0.1%) for 20 min at room temperature with gentle rocking. Fixed brains were washed 3 times for 15 min with PBT (PBS and Triton X-100 0.3%) and then incubated in PBT-BSA for at least 1h before incubation with the corresponding primary antibody/antibodies. The following primary antibodies were used in this study: guinea pig anti-Dpn (1:2,000; [42]), rabbit anti-Ase (1:100; [42]), goat anti-Numb (1:200; Santa Cruz Biotechnology, sc-23579), and rabbit anti-PKCζ (1:100; Santa Cruz Biotechnology, sc-216). Fluorescence images corresponding to Figure 1 and Figure 2a,b were recorded using an Inverted Leica laser-scanning spectral confocal microscope TCS SP2. Fluorescence images in Figure 2c and Figure 6b were recorded using a Super-resolution Inverted Confocal Microscope Zeiss LSM 880-Airyscan Elyra PS.1 (Figure 2c) or an Inverted Confocal Microscope Olympus FV1200 (Figure 6b), respectively.

### 4.3. Statistics

Data related to the ACD regulator localization failures were analyzed with a chi-squared test (with a Yates correction). The sample size (n) and the *p*-value are indicated in the figure or figure legend; * *p* < 0.05, ns: not significant (*p* > 0.05).

## Figures and Tables

**Figure 1 ijms-22-11332-f001:**
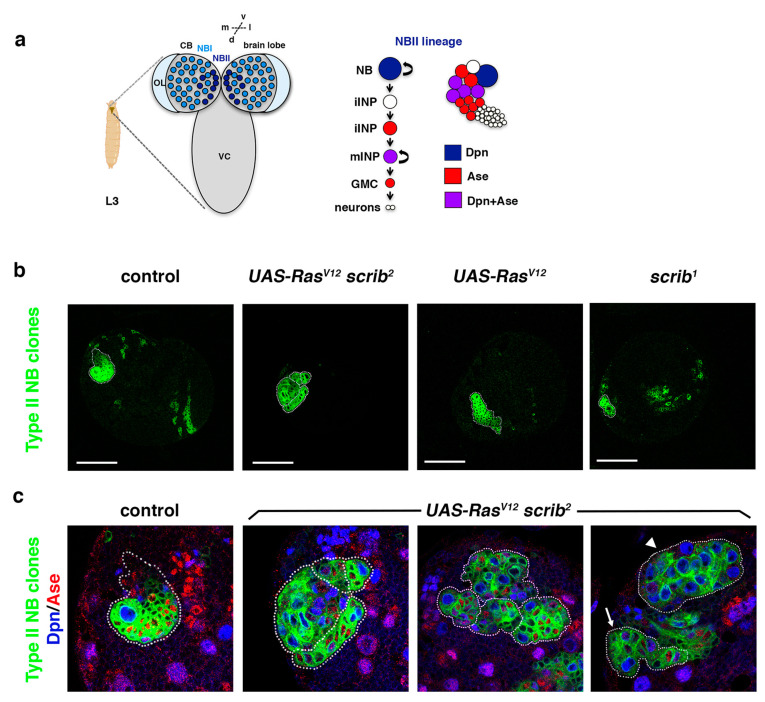
*Ras^V^*^12^*scrib* NBII mutant clones do not show tumor-like overgrowth. (**a**) Type II neuroblast (NBII) lineages (8 per brain hemisphere) are found at particular locations at the dorsomedial part of each larval brain hemisphere, whereas type I NB (NBI) lineages are spread over the central brain (CB); L3: third instar larva; OL: optic lobe; VC: ventral cord; d: dorsal; v: ventral; m: medial; l: lateral. In NBII lineages, the NB divides to give rise to an intermediate neural progenitor (INP) that, after a maturation process, divides to generate another INP and a ganglion mother cell (GMC) that will terminally divide to give rise to two different neurons (or glial cells). The NB in NBII lineages expresses the transcription factor Dpn, whereas mature INPs (mINPs) express both transcription factors Dpn and Ase; iINP (immature INP). (**b**) Confocal micrographs showing a brain hemisphere with NBII lineages of the indicated genotypes. *scrib*^1^ NBII null mutant clones are smaller than control clones, whereas *Ras^V^*^12^
*scrib*^2^ NBII mutant clones show variable sizes as represented in (**c**); *Ras^V^*^12^ NBII mutant clones are similar to control clones. (**c**) Confocal micrographs showing NBII lineages of the indicated genotypes stained with Dpn (blue) and Ase (red), all at the same magnification; most *Ras^V^*^12^
*scrib*^2^ NBII mutant clones are smaller than control clones, with ectopic NBs (in blue; Dpn^+^ Ase^−^), and they appear frequently in groups (dotted lines delimitate each NBII clone); some *Ras^V^*^12^
*scrib*^2^ NBII mutant clones are similar to control clones (arrow) and a few of them were composed mainly of NBs (arrowhead), but none of them showed tumor-like overgrowth. In both (**b**,**c**), the *Dll-Gal4 UAS-CD8::GFP;*
*FRT82B tub-Gal80* line was used to perform MARCM clones [44] in type II NBs; scale bar: 50 μm.

**Figure 2 ijms-22-11332-f002:**
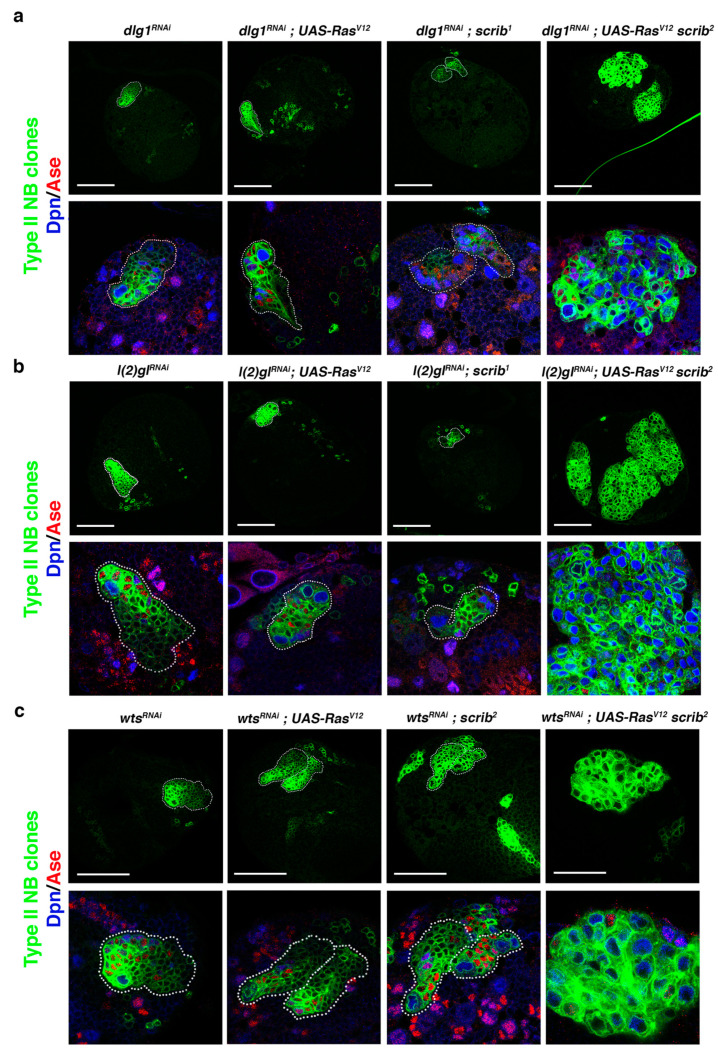
Downregulation of known ACD regulators in *Ras^V^*^12^
*scrib* NBII mutant clones induces tumor-like overgrowth. (**a**) Confocal micrographs showing brain hemispheres with NBII lineages of the indicated genotypes. Below each hemisphere, detailed views of the corresponding NBII lineage stained with Dpn (blue) and Ase (red) are displayed at the same magnification. Some *dlg1^RNAi^; Ras^V^*^12^
*scrib*^2^ NBII clones, as the clone shown, present tumor-like overgrowth, with the clone filled mainly by NBs (in blue; Dpn^+^Ase^−^), whereas the other genetic combinations never show tumor-like overgrowth. (**b**) Confocal micrographs showing brain hemispheres with NBII lineages of the indicated genotypes. Below each hemisphere, detailed views of the corresponding NBII lineage stained with Dpn and Ase are displayed at the same magnification. Only *l(2)gl^RNAi^; Ras^V^*^12^
*scrib*^2^ NBII clones show tumor-like overgrowth, tumoral masses filled mainly by ectopic NBs (in blue; Dpn^+^Ase^−^). (**c**) Confocal micrographs showing brain hemispheres with NBII lineages of the indicated genotypes. Below each hemisphere, detailed views of the corresponding NBII lineage stained with Dpn and Ase are displayed at the same magnification. Only *wts^RNAi^; Ras^V^*^12^
*scrib*^2^ NBII clones show tumor-like overgrowth, tumoral masses filled mainly by ectopic NBs (in blue; Dpn^+^Ase^−^); scale bar: 50 μm.

**Figure 3 ijms-22-11332-f003:**
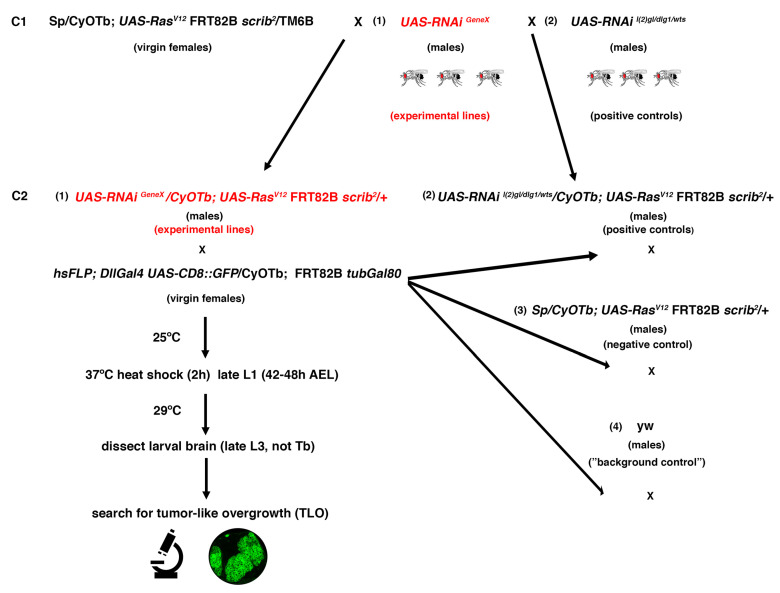
Outline of the crossing scheme and workflow for the RNAi screen. C1: Crosses 1; C2: Crosses 2; L1: first instar larvae; L3: third instar larvae; AEL: after egg laying; TLO: tumor-like overgrowth.

**Figure 4 ijms-22-11332-f004:**
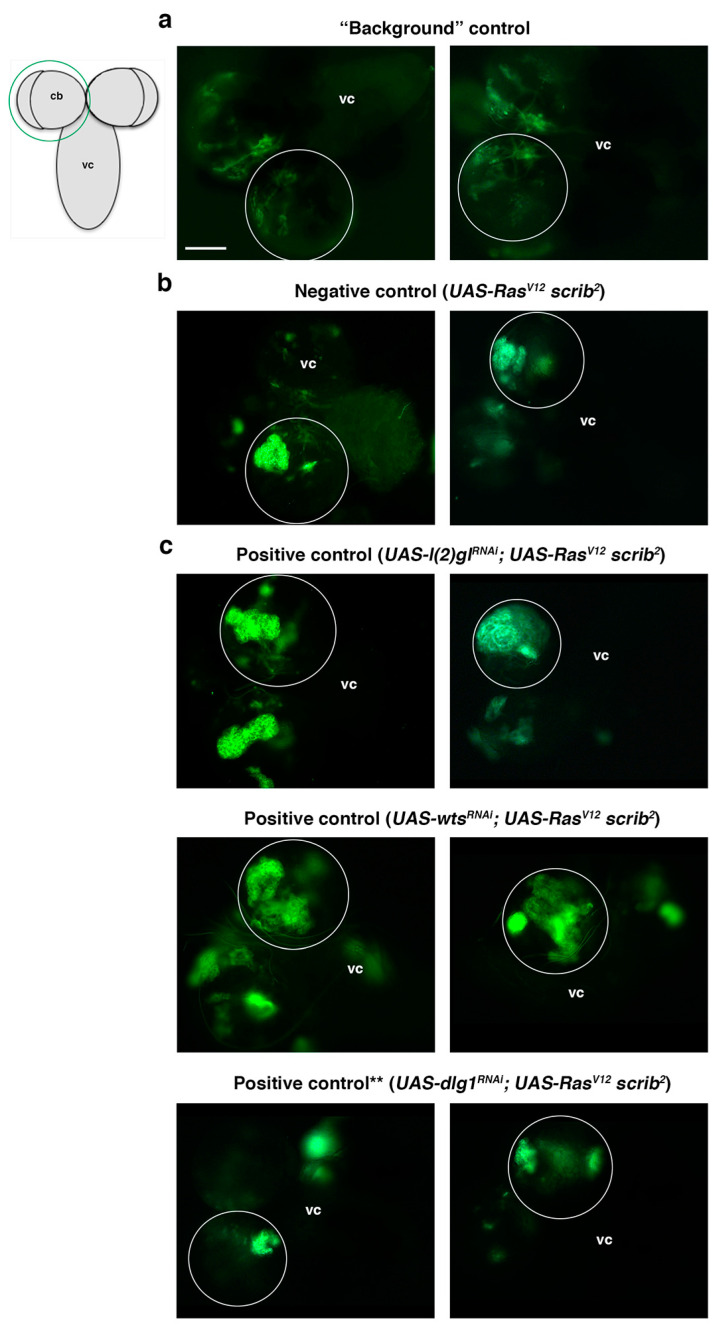
Screen controls. All panels show a dorsal view of a larval CNS that includes the ventral cord (vc) and the two brain hemispheres, one of which is encircled, as represented in the schematic larval CNS; cb (central brain). (**a**) A background control in which the GFP detected is leaky GFP or autofluorescence, as the Gal80 repressor is present to inhibit the CD8::GFP expression driven by the Gal4 line (see also Figure 3). (**b**) Two examples of the biggest *Ras^V^*^12^
*scrib*^2^ NBII clones found, which is our established “negative control” (i.e., not considered TLO). Clones in the experimental lines above that size are considered TLO and potential “positive lines.” (**c**) Positive controls, which are known ACD regulators, including *l(2)gl* and *wts*, whose downregulation in *Ras^V^*^12^
*scrib*^2^ NBII clones induce TLO; (**) The downregulation of *dlg1*, another potential positive control, in *Ras^V^*^12^
*scrib*^2^ NBII clones does not show clear TLO when tested under the conditions of the screen workflow (see also text and Table 1). Scale bar: 100 μm for all panels.

**Figure 5 ijms-22-11332-f005:**
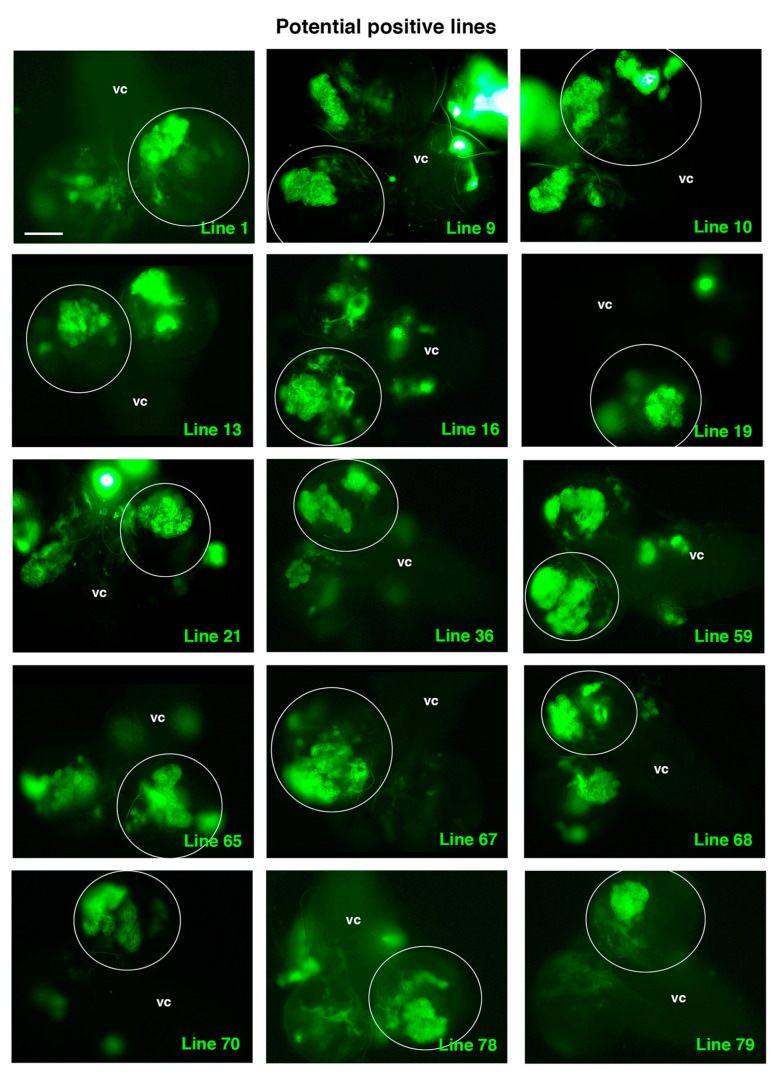
Potential positive lines selected. All panels show a dorsal view of a larval CNS that includes the ventral cord (vc) and the two-central brain (cb) hemispheres, one of which is encircled. All the selected experimental *UAS-RNAi* lines shown present clones with TLO and were considered potential positive lines following the established criteria (see text and Table 1). Scale bar: 100 μm for all panels.

**Figure 6 ijms-22-11332-f006:**
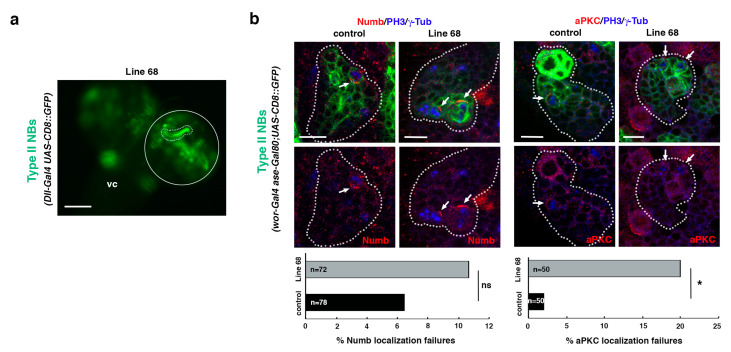
Line 68 is a potential ACD regulator. (**a**) Dorsal view of line 68 larval CNS, which includes the ventral cord (vc) and the two-central brain (cb) hemispheres, one of which is encircled. One NBII MARCM clone is shown surrounded by a dotted line. Scale bar: 100 μm. “Line 68” represents the genotype: *hsFLP; Dll-Gal4 UAS-CD8::GFP/UAS-RNAi^68^; FRT82B/FRT82B* (mutant clone) with *hsFLP; Dll-Gal4 UAS-CD8::GFP/UAS-RNAi^68^; FRT82B tubGal80/FRT82B tubGal80* (twin wild-type clone, not labeled), all in an *hsFLP; Dll-Gal4 UAS-CD8::GFP/UAS-RNAi^68^; FRT82B tubGal80/FRT82B* background. (**b**) Confocal micrographs of control and line 68 NBII lineages. The downregulation of the gene associated with line 68 induces significant failures in the localization of aPKC, whereas no significant defects in the localization of Numb are detected. In all panels, PH3 labels mitotic cells and γ-Tub the centrosomes. White arrows indicate the crescent correctly formed in metaphase progenitors in control clones and the absence of the aPKC crescent in the mutant condition. Data were analyzed with a chi-squared test (Yates correction), * *p* < 0.05 (*p* = 0.011) and ns: not significant (*p* > 0.05); n depicts the number of dividing cells analyzed; scale bars: 10 μm. “Control” corresponds to the genotype: *wor-Gal4 aseGal80/wor-Gal4 aseGal80*; *UAS-CD8::GFP/UAS-CD8::GFP*, and “Line 68” represents the genotype: *wor-Gal4 aseGal80/UAS-RNAi^68^; UAS-CD8::GFP/+*.

**Table 1 ijms-22-11332-t001:** Control and *UAS-RNAi* lines were analyzed on the screen. Background, negative and positive controls were run (see Figure 3 for detailed genotypes). *dlg1^RNAi^*; *Ras^V^*^12^
*scrib*^2^ larval brains did not show clear cases of TLO in the larvae analyzed (see also text). An additional control for the KK library landing site (LS) at 40D, without any RNAi line, was carried out, as the *UAS-wts^RNAi^* line was inserted at that location. Fifteen potential positive lines (highlighted in green), i.e., those that showed TLO following the established criteria (see text for details), were selected out of 79 *UAS-RNAi* lines screened, which finally corresponded with 77 different genes (as lines 47 and 48 represent the same gene, as well as lines 61 and 75). Lines 28, 37, 38, 49, and 50 are currently discarded in VDRC.

Controls	Genotype	# Larvae Dissected	# Larvaewith Clones	# Larvae with TLO Clones	% Larvae with TLO Clones	VDRC ID	OFF Targets	CG Number	Gene Symbol
Background	*yw*	30	0	0	0.0%				
Negative	*Ras^V12^ scrib^2^*	93	35	0	0.0%				
Positives:	*l(2)gl^RNAi^; Ras^V12^ scrib^2^*	58	21	8	**38.0%**				
	*wts^RNAi^; Ras^V12^ scrib^2^*	70	40	6	**15.0%**				
	*dlg1^RNAi^; Ras^V12^ scrib^2^*	42	16	0	0.0%				
Control LS	*40D-UAS; Ras^V12^ scrib^2^*	34	13	0	0.0%				
**RNAi LINES:**	**1**	77	27	2	**7.4%**	105852/KK	0	CG8815	*Sin3A*
	**2**	27	13	0	0.0%	104803/KK	0	CG4336	*rux*
	**3**	47	21	0	0.0%	104829/KK	0	CG10756	*Taf13*
	**4**	44	16	0	0.0%	105478/KK	0	CG44247	*CG44247*
	**5**	49	17	0	0.0%	105384/KK	0	CG6093	*abo*
	**6**	47	16	0	0.0%	105462/KK	0	CG8428	*spin*
	**7**	60	27	0	0.0%	104335/KK	0	CG2917	*Orc4*
	**8**	45	12	0	0.0%	105502/KK	1	CG5216	*Sirt1*
	**9**	63	23	**2**	**8.7%**	104496/KK	0	CG17870	*14.3.3* *𝜁*
	**10**	51	20	**2**	**10.0 %**	105409/KK	0	CG5343	*Bug22*
	**11**	22	12	0	0.0%	105367/KK	0	CG1616	*dpa*
	**12**	37	16	0	0,0%	105501/KK	2	CG5271	*RpS27A*
	**13**	18	15	**2**	**13.3%**	103716/KK	0	CG4088	*Orc3*
	**14**	34	15	0	0.0%	106526/KK	0	CG13403	*CG13403*
	**15**	32	16	0	0.0%	106688/KK	1	CG5193	*TfIIB*
	**16**	46	19	**3**	**15.8%**	109108/KK	0	CG12559	*rl*
	**17**	30	12	0	0.0%	106185/KK	0	CG10052	*Rx*
	**18**	24	15	0	0.0%	106153/KK	0	CG2914	*Ets21C*
	**19**	50	19	**3**	**15.8%**	108828/KK	2	CG18497	*spen*
	**20**	32	12	0	0.0%	107026/KK	0	CG31739	*AspRS-m*
	**21**	30	18	**2**	**11.1%**	105739/KK	0	CG3291	*pcm*
	**22**	33	14	0	0.0%	106142/KK	0	CG8817	*lilli*
	**23**	23	13	0	0.0%	106196/KK	0	CG9576	*Phf7*
	**24**	59	13	0	0.0%	34113/GD	1	CG4494	*smt3*
	**25**	36	19	0	0.0%	32889/GD	0	CG1736	*Prosα3T*
	**26**	45	14	0	0.0%	1603/GD	2	CG3066	*Sp7*
	**27**	34	13	0	0.0%	35061/GD	0	CG6061	*mip120*
	**28**	28	12	0	0.0%	27424/GD	104	CG43398	*scrib*
	**29**	25	13	0	0.0%	34210/GD	1	CG8023	*eIF4E3*
	**30**	17	16	0	0.0%	30587/GD	0	CG3886	*Psc*
	**31**	28	12	0	0.0%	27467/GD	1	CG5604	*Ufd4*
	**32**	25	12	0	0.0%	9396/GD	0	CG3352	*ft*
	**33**	29	14	0	0.0%	105948/KK	0	CG40486	*CG40486*
	**34**	25	16	0	0.0%	2919/GD	0	CG9653	*brk*
	**35**	34	13	0	0.0%	25387/GD	0	CG1977	*α-Spec*
	**36**	38	16	4	**25.0%**	105471/KK	2	CG2577	*CG2577*
	**37**	33	12	0	0.0%	16331/GD	1	CG42616	*Cul3*
	**38**	21	12	0	0.0%	32652/GD	2	CG15835	*Kdm4A*
	**39**	25	12	0	0.0%	35709/GD	0	CG16799	*CG16799*
	**40**	34	12	0	0.0%	3122/GD	0	CG17610	*grk*
	**41**	49	15	0	0.0%	38233/GD	1	CG43758	*sli*
	**42**	36	13	0	0.0%	12965/GD	1	CG17280	*levy*
	**43**	23	12	0	0.0%	25344/GD	0	CG1848	*LIMK1*
	**44**	20	12	0	0.0%	25549/GD	0	CG7762	*Rpn1*
	**45**	28	12	0	0.0%	30586/GD	0	CG3886	*Psc*
	**46**	28	13	0	0.0%	26888/GD	0	CG7771	*sim*
	**47**	44	16	0	0.0%	2947/GD	0	CG10798	*Myc*
	**48**	59	16	0	0.0%	2948/GD	0	CG10798	*Myc*
	**49**	30	12	0	0.0%	36086/GD	0	CG9124	*eIF3h*
	**50**	21	12	0	0.0%	16381/GD	0	CG12000	*Prosβ7*
	**51**	23	16	0	0.0%	106071/KK	0	CG14226	*dome*
	**52**	33	13	0	0.0%	106155/KK	3	CG10325	*abd-A*
	**53**	45	29	0	0.0%	103619/KK	2	CG7538	*Mcm2*
	**54**	29	18	0	0.0%	106459/KK	1	CG1716	*Set2*
	**55**	28	13	0	0.0%	105865/KK	0	CG11158	*CG11158*
	**56**	21	12	0	0.0%	104415/KK	0	CG1354	*CG1354*
	**57**	29	13	0	0.0%	105494/KK	0	CG4400	*Brms1*
	**58**	25	13	0	0.0%	102054/KK	1	CG8367	*cg*
	**59**	25	17	**2**	**11.8%**	104775/KK	0	CG9907	*para*
	**60**	17	13	0	0.0%	106542/KK	0	CG14817	*CG14817*
	**61**	41	16	0	0.0%	2915/GD	4	CG5055	*baz*
	**62**	26	15	0	0.0%	106449/KK	0	CG2272	*slpr*
	**63**	34	13	0	0.0%	105371/KK	0	CG17437	*wds*
	**64**	43	22	0	0.0%	104753/KK	1	CG10445	*CG10445*
	**65**	43	23	**2**	8.7%	105946/KK	1	CG12238	*e(y)3*
	**66**	53	33	0	0.0%	106505/KK	0	CG12728	*CG12728*
	**67**	47	21	**3**	**14.3%**	106503/KK	0	CG1561	*pkm*
	**68**	30	14	**3**	**21.4%**	104425/KK	0	CG7846	*Arp8*
	**69**	23	14	0	0.0%	104770/KK	0	CG15865	*CG15865*
	**70**	32	13	2	**15.4%**	105374/KK	1	CG11734	*HERC2*
	**71**	53	30	1	3.3%	104792/KK	0	CG33980	*Vsx2*
	**72**	31	13	0	0.0%	21867/GD	0	CG4547	*Atx-1*
	**73**	38	14	0	0.0%	104427/KK	2	CG32697	*Ptpmeg2*
	**74**	30	15	0	0.0%	106491/KK	1	CG4320	*raptor*
	**75**	32	13	0	0.0%	2914/GD	4	CG5055	*baz*
	**76**	26	13	0	0.0%	104963/KK	1	CG33323	*Fer1*
	**77**	27	15	0	0.0%	104600/KK	0	CG42267	*RunxB*
	**78**	35	13	2	**15.4%**	105942/KK	0	CG7280	*shop*
	**79**	31	17	2	**11.8%**	105509/KK	0	CG1803	*regucalcin*

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
