# Peer review of "Pilot RNAi Screen in Drosophila Neural Stem Cell Lineages to Identify Novel Tumor Suppressor Genes Involved in Asymmetric Cell Division"

_ijms, 2021, doi:10.3390/ijms222111332_

Round 1
Reviewer 1 Report
In the manuscript by Sandra Manzanero-Ortiz, the authors claim to have devised an ingenious screen to identify new and potential modifiers of asymmetric cell division in the drosophila neuroblast lineage.
The manuscript completely fails to substantiate its aims.
In the entire manuscript no measurement of ACD is provided, not even to check whether it is impaired in their so called positive controls.
Only in figure 6 the authors show some mis-localization of a protein involved in ACD, yet no measurement of ACD itself.
Telophase rescue is a known phenomenon in ACD and therefore mis-localization of aPKC does not reflect defects in ACD.
Essentially none of the authors conclusions are substantiated by the data presented as they do not provide any quantification (just visual comparison without statement on animals observed).
The text is littered with statements that are not supported by the data:
115 “mutant clones appeared with low frequency and were of small size compared with control”
116 “clones were detected at the same frequency”
117 “their size were variable”
132 “some brains”
133 “filled mainly by NBs”
152 “observing big”
173 “similar to those”
Etc…
Frequency, size, similarity, NB/INP content is not measured in any of the conditions reported.
The authors have to include quantifications for all their observations to substantiate their statements.
Quantifications of clone size and volume accompanied by proper statistics and indication of animals used.
Quantifications of Dpn+/Ase- vs Ase+ cells inside clones accompanied by proper statistics and indication of animals used.
Quantifications of frequency, eg. number of clones in animals and how many animals have clones.
This is indeed problematic as the authors state line 219 “The downregulation of dlg1, another potential positive control, in RasV12 scrib2 NBII clones does not show clear TLO when tested under the conditions of the screen workflow (see also text and Table 1)”.
But the authors claim earlier in the text it is a perfect example, line 132 Intriguingly, we observed some brains with dlg1RNAi; RasV12 scrib NBII clones bigger than RasV12 scrib clones and filled mainly by NBs (Dpn+ Ase-) (Figure 2a), a phenotype that also appeared and much more exacerbated in brains with l(2)glRNAi; RasV12 scrib NBII clones, which expanded in some cases throughout the brain hemisphere (Figure 2b).
The authors inappropriately represent the data, as their magnifications, presented in the figure panels on the bottom are not the same size as in the top panels for all figures.
Hence this is obvious for Figure 1C.
The crop of wtsRNAi; RasV12 scrib2 is not the same as other panels.
All panels should be accompanied with scale bars (including figure 4, 4, 6a where they are completely missing).
Related to their screen, the MARCM is not properly cited, nor do the authors provide a clear definition of what a TLO is (due to a lack of quantitative data, the authors should use Imaris or other 3D software to measure the volume of GFP positive clones).
The authors use two Scrib alleles, scrib1 or scrib2, without stating why and whether they are the same.. (quantification needed to justify this).
The authors declare no conflict of interest, yet fail to include proper identifiers for their RNAi lines tested (hence the targets are unknown).
All VDRC identifiers should be included in Table S1
In addition, just out of curiosity….
One picks 79 RNAi lines randomly.. and 15 show positive hit for TLO (assumed ACD)..
This means that if one would screen the drosophila genome close two 20% of all genes are involved in ACD. The authors do not discuss this aspect.
Many Ras sensitized screens have been reported and the authors fail to cite these studies.
This is a poor manuscript that reads as if it were part of the methods section of a PhD report and should be rejected as such.
Author Response
Please, see the attachment

Reviewer 2 Report
Manzanero-Ortiz et al., presented convincing results on their pilot study in which they isolated new tumor suppressor genes involved in the regulation of ACD. These study provides a solid basis to perform a more robust screening and deeper molecular analyses. The manuscript is well written and deserves publication in the present form.
Author Response
Thank you
Reviewer 3 Report
In their manuscript entitled “Pilot RNAi screen in Drosophila neural stem cell lineages to identify novel tumor suppressor genes involved in asymmetric cell division,” Manzanero-Ortiz present a pilot screen to identify RNAi constructs that result in overgrowth of RasV12, scrib mutant clones in the Drosophila larval brain. I find their approach clever and promising, and it is a very nice example of how a sensitized screen can reveal novel players in a signaling pathway and what potential a full screen might offer. The phenotypes and data in the figures are sound and presented well. However, as written this is the single advance that this manuscript reports, that it is possible to do a sensitized screen and discover new components regulating ACD and connected to tumorigenesis. We also have to take this claim at face value, as there is zero information on what was actually screened. This makes the relevance of this paper, at best, targeted to a narrow handful of researchers.
Major comments:
- At the bare minimum, the identity of RNAi Line 68, which is characterized in some detail, must be provided as well as a full list of RNAi lines that were screened. While my personal opinion is that Table 1 should include this information, I respect that due to competition in the field the authors might not want to provide this information. However, the authors could provide a supplementary table with the RNAi line stock numbers and target genes assembled in a random list that is not correlated with the phenotypes reported in Table 1. For some general background on the discussion around what information should be provided with published RNAi screen data (for example the MIARE standards) and why, please see http://miare.sourceforge.net/HomePage , doi: 1002/wrna.110 or https://www.ddw-online.com/a-decade-of-rnai-screening-too-much-hay-and-very-few-needles-676-201308/.
- Text line 159: The authors need to include how many genes are represented by these 79 hairpins. To judge the validity of the RNAi pilot screen, it would also be helpful to know if more than 1 hairpin was identified per gene. Two independent hairpins producing the same phenotype ensures that the result is not simply an off-target effect. It is promising that known ACD modifiers turned-up in the screen, but there is no way to judge if the authors identified anything new and if new genes might be real possible players in this process.
- Please briefly explain in the methods the logic used to select the genes targeted by the RNAi hairpins. ACD is a very specific process and based on my own lab’s experience with RNAi screens, I question if 15 out of 79 truly “random” hairpins (~20%!) would show such a tumor phenotype.
- The observed rates of TLO clones are overall quite low. L(2)gl at almost 40% gives a reasonably strong phenotype, but can the authors comment on what TLO clone rates around 10% mean? TLO clones aren’t observed at all in the control, of course, but do the low TLO rates mean that the identified genes have minor roles in the process, are easily compensated, or something else?
Additional comments:
- Can the authors add a line or additional labels to Figure 1c. I think the first panel is control and the other 3 panels are RasV12, scrib clones, but I am not sure.
- Figure 2c – the warts clone is larger, but it doesn’t look like there are necessarily more cells, just that Ase is lost and the Dpn+ cells are way bigger in size, while the images for dlg1 and l(2)gl appear to really have more cells. If the cells are just bigger, is this really a tumor? Also, are the blow-up images all the same magnification? I assume so but there are no scale bars to confirm, and this is important when comparing clone size across the images.
- Text Line 243, how do the authors address off-target effects in their screen? This is the first mention. Reporting how many RNAi lines targeted the same gene and if more than a single hairpin was found per gene would help convince the reader the observed TLO effects are not just off-target or background issues.
- Please include the full genotype of the control in Figure 6a and 6b in the legend.
Round 2
Reviewer 1 Report
As the authors do not provide quantitative data associated with statistics, I do not feel the conclusions of the manuscript are supported by the data.
Reviewer 3 Report
The authors have sufficiently addressed all of my comments on the first version of the manuscript. The new information in the table and textual modifications have greatly improved accessibility of the paper and the scientific quality, and have not raised any additional concerns. There are a few grammar errors in the new modifications to the text, but they are minor.
Author Response
We would like to thank the reviewer for all his/her input. We have now corrected those few grammar errors we have detected in the text (i.e. "mutant" instead of "muta" in the Figure 6 legend; "Bazooka" instead of "Bazzoka" in the Conclusions paragraph (as well as a couple of other minor things).
Round 3
Reviewer 1 Report
.